# Beyond Euclidean: Dual-Space Representation Learning for Weakly Supervised Video Violence Detection

**Jiaxu Leng**[1,2], **Zhanjie Wu**[1,2], **Mingpi Tan**[1,2], **Yiran Liu**[1], **Ji Gan**[1,2],
**Haosheng Chen**[1,2], **Xinbo Gao** [1,2] *

[1] Chongqing University of Posts and Telecommunications, Chongqing, China
[2] Chongqing Institute for Brain and Intelligence, Guangyang Bay Laboratory, Chongqing, China
gaoxb@cqupt.edu.cn

## Abstract

While numerous Video Violence Detection (VVD) methods have focused on representation learning in Euclidean space, they struggle to learn sufficiently discriminative features, leading to weaknesses in recognizing normal events that are visually similar to violent events (*i.e.*, ambiguous violence). In contrast, hyperbolic representation learning, renowned for its ability to model hierarchical and complex relationships between events, has the potential to amplify the discrimination between visually similar events. Inspired by these, we develop a novel Dual-Space Representation Learning (DSRL) method for weakly supervised VVD to utilize the strength of both Euclidean and hyperbolic geometries, capturing the visual features of events while also exploring the intrinsic relations between events, thereby enhancing the discriminative capacity of the features. DSRL employs a novel information aggregation strategy to progressively learn event context in hyperbolic spaces, which selects aggregation nodes through layer-sensitive hyperbolic association degrees constrained by hyperbolic Dirichlet energy. Furthermore, DSRL attempts to break the cyber-balkanization of different spaces, utilizing cross-space attention to facilitate information interactions between Euclidean and hyperbolic space to capture better discriminative features for final violence detection. Comprehensive experiments demonstrate the effectiveness of our proposed DSRL.

## 1 Introduction

Compared with manually checking out violent events in surveillance videos which is time-consuming and laborious, Video Violence Detection (VVD), which focuses on identifying violent events in videos and provides automatic and instantaneous responses, has gained significant research attention due to its potential applications. However, it is expensive to annotate each frame in a video so that we can train a VVD model with supervised learning. To address this, current methods often utilize weakly supervised settings to formulate the problem as a multiple-instance learning (MIL)[19] task. These methods treat a video as a bag of instances (*i.e.,* snippets or segments), and predict their labels based on the video-level annotations.

According to the modality type of the input data, existing weakly supervised VVD methods can be roughly divided into two categories, unimodal with only vision input and multimodal with vision and audio input. The unimodal methods [29, 36, 8, 10, 20] learn the different distributions of normal and violent events through video-level labels, focus on finding valuable visual cues that are distinct from non-violent events and use them to detect violent events, *i.e.,* fighting. However, relying on visual cues

---

*Corresponding author

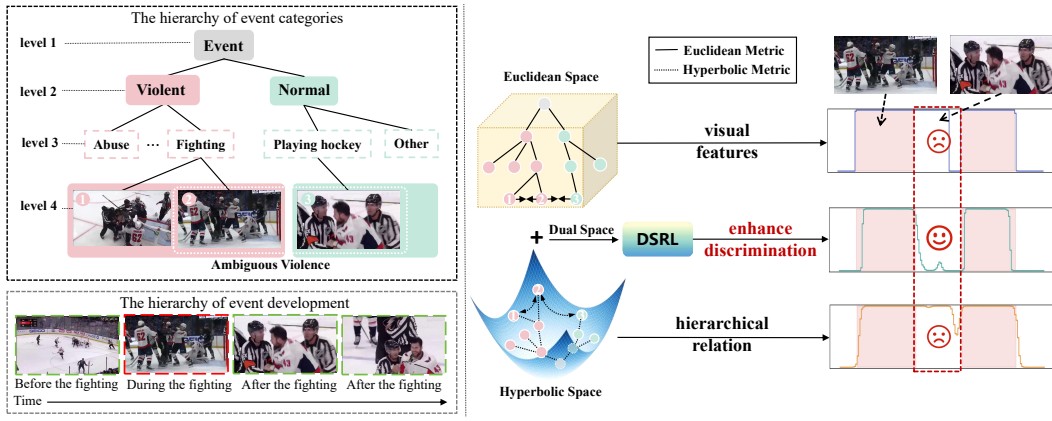

| (a) Hierarchical Diagram in VVD | (b) Superiority of DSRL for Ambiguous Violence Detection |

Figure 1: (a) Hierarchical diagram in Video Violence Detection (VVD). (b) Our DSRL enhances the detection of ambiguous violence by combining Euclidean and Hyperbolic spaces to balance visual feature expression and hierarchical event relations.

to identify violent events is sometimes unreliable, especially when facing visually ambiguous events, like normal physical collisions and fighting behavior in hockey games. To alleviate this problem, Wu et al [34] released a large multimodal violence dataset, named XD-Violence, and accelerated a series of multimodal VVD methods [34, 35, 22, 37, 23]. Multimodal VVD methods incorporate not only visual cues but also complementary audio information for improving the discrimination of violent events. Previous methods have employed Euclidean space representation learning and achieved good results in many other computer vision tasks[27, 28, 15, 32]. Despite advancements in audio-visual violent video detection (VVD) methods, their performance remains unsatisfactory in recognizing normal events that are visually similar to violent events (i.e., ambiguous violence) due to the limitations of Euclidean space. While visual features are fully extracted in Euclidean space, these methods fail to adequately capture and utilize the intrinsic relations between events.

To fully understand an event, on the one hand, we need to explore the hierarchy of event categories, on the other hand, we need to sort through the events, including the trend before the event, the action during the event and the behaviour after the event, which reflects the hierarchy of event development, shown in Figure 1(a). An ambiguous violent event is confusing at the current category level or happening moment, but may easily be detected if we can grasp the hierarchical relations. Fortunately, hyperbolic representation learning, characterized by exponentially increasing the metric distances and naturally reflects the hierarchical structure of data, has gained attention and shown promising performance in computer vision tasks, like semantic segmentation [1], medical image recognition [38], action recognition [24, 17], anomaly recognition [12]. At present, only one method, HyperVD [25], makes a preliminary attempt on the VVD task via hyperbolic representation learning. HyperVD introduces the Hyperbolic Graph Convolutional Networks (HGCN) [3], an extended version of Euclidean graph convolution for representation learning in hyperbolic space, to learn discriminative representations. However, HGCN employs a hard node selection strategy during the message passing, where the nodes whose correlation is higher than the threshold (a manual parameter) are selected for message aggregation and otherwise discarded, which leads to insufficient hierarchical relation learning. In addition, existing VVD methods deploy feature embedding either in Euclidean or hyperbolic spaces. Representation learning in a single space is like picking the sesame and losing the watermelon, where the feature embedding is insufficient to guarantee the performance of VVD. On the one hand, hyperbolic representation learning strengthens the hierarchical relation of events but weakens the expression of visual features, on the other hand, Euclidean representations emphasize visual features but ignore relations between events. Therefore, leveraging the advantages of both spaces is essential for improving the performance of VVD methods, shown in Figure 1(b).

In this paper, we propose a novel Dual-Space Representation Learning (DSRL) method for weakly supervised VVD under the multimodal input setting. Specifically, we designed two customized modules, the Hyperbolic Energy-constrained Graph Convolutional Network module (HE-GCN) and the Dual-Space Interaction module (DSI). Instead of adopting the hard node selection strategy in HGCN, HE-GCN selects nodes for message aggregation by our introduced layer-sensitive hyperbolic association degrees, which are dynamic thresholds determined by the message aggregation degree at

each layer. To better align with the characteristics of hyperbolic spaces, we introduce the hyperbolic Dirichlet energy to quantify the extent of message aggregation. Benefiting from the dynamic threshold, the layer-by-layer focused message passing strategy adopted by HE-GCN not only ensures the efficiency of information excavation but also improves the model's comprehensive understanding of the events, thus enhancing the model's ability to discriminate ambiguous violent events. Although hyperbolic representation learning enhances the understanding of hierarchical relations of events, the role of visual representations in violence detection cannot be discarded. However, fusing representation in different spaces remains a challenge, to break the information cocoon, DSI utilises cross-space attention to facilitate information interactions between Euclidean and hyperbolic space to capture better discriminative features, where Euclidean representations have effectiveness on the significant motion and shape changes in the video, while hyperbolic representations accelerate the comprehension of hierarchical relations between events, working together to improve the performance of violence detection in videos.

**Contributions**: **(1)** To the best of our knowledge, our DSRL is the first method to integrate Euclidean and hyperbolic geometries for VVD, significantly improving discrimination of ambiguous violence and achieving state-of-the-art performance on the XD-Violence dataset in both unimodal and multimodal settings. **(2)** To better capture the hierarchical context of events, we design the HE-GCN module with a novel message aggregation strategy, where the node selection threshold is dynamic not fixed and determined by layer-sensitive hyperbolic association degrees based on hyperbolic Dirichlet energy. **(3)** To break the information cocoon for better dual-space cooperation, visual discrimination from Euclidean and event hierarchical discrimination from hyperbolic, we propose the DSI module, which utilizes cross-space attention to facilitate information interactions.

# 2   Related Work

**Weakly Supervised Video Violence Detection.** Weakly supervised VVD requires identifying violent snippets under video-level labels, where the MIL [19] framework is widely used for denoising irrelevant information. Recently, progress has been made in weakly supervised VVD, with approaches categorized into two main categories: vision-based and audio-visual-based methods. Employing exclusively visual cues, vision-based VVD endeavours to discern the occurrence of violent events within videos. Most existing works [7, 26, 30, 33] consider VVD as solely a visual task, and CNN-based networks are utilized to encode visual features. However, these approaches overlook the interaction between different modalities and the relevant audio information, which could negatively impact the accuracy of violence detection. To alleviate this problem, Wu et al [34] released a large multimodal violence dataset, named XD-Violence, and accelerated a series of multimodal VVD methods [34, 35, 22, 37, 23, 18, 25]. In contrast to unimodal methods, they incorporate not only visual cues but also complementary audio information for improving the discrimination of ambiguous violent events. Subsequently, many studies [22, 23] have focused on the integration of visual and audio information. There is also work[37] focused on solving the modality asynchrony problem. Despite the progress of weakly supervised multimodal VVD methods, all the above methods carry out representation learning in Euclidean Spaces, making it difficult to effectively handle ambiguous violence.

**Hyperbolic Representation Learning.** Hyperbolic representation learning, characterized by exponentially increasing the metric distances and naturally reflects the hierarchical structure of data, has gained attention and shown promising performance in computer vision tasks, like semantic segmentation [1], visual representation learning [9], medical image recognition [38], action recognition [24, 17], anomaly recognition [12]. More Recently, HyperVD [25] has made an initial attempt at the VVD task using hyperbolic representation learning. HyperVD incorporates Hyperbolic Graph Convolutional Networks (HGCN) [3] to acquire discriminative representations. However, while hyperbolic representation learning enhances the hierarchical relationships of events, it diminishes the representation of visual features. Therefore, we propose a novel Dual-Space Representation Learning to joint the strengths of Euclidean and hyperbolic space. Meanwhile, we design HE-GCN to gradually shift its focus from capturing global contextual information to concentrating on crucial detailed features, progressively capturing the hierarchical context of events.

# 3   Preliminaries

**Problem Definition.** Given an video sequence $S = \{S_t\}_{t=1}^T$ with $T$ non-overlapping segments. For a video segment, the weakly supervised VVD requires distinguishing whether it contains violent events via an events relevance label $y_t \in \{0, 1\}$, where $y_t = 1$ means in the current segment includes violent cues.

**Hyperbolic Geometry.** A Riemannian manifold $(\mathcal{M}, g)$ of dimension $n$ is a real and smooth manifold equipped with an inner product on tangent space $g_x \colon \mathcal{T}_x\mathcal{M} \times \mathcal{T}_x\mathcal{M} \to \mathbb{R}$ at each point $x \in \mathcal{M}$, where the tangent space $\mathcal{T}_x\mathcal{M}$ is a $n$-dimensional vector space and can be seen as a first-order local approximation of $\mathcal{M}$ around point $x$. In particular, hyperbolic space $(\mathbb{D}_c^n, g^c)$, a constant negative curvature Riemannian manifold, is defined by the manifold $\mathbb{D}_c^n = \{x \in \mathbb{R}^n : c \|x\| < 1\}$ equipped with the following Riemannian metric: $g_x^c = \lambda_x^2 g^E$, where $\lambda_x := \frac{2}{1-c\|x\|^2}$ and $g^E = I_n$ is the Euclidean metric tensor. Considering the numerical stability and calculation simplicity of its exponential and logarithmic maps and distance functions, we select the Lorentz model [21] as the framework cornerstone.

**Lorentz Model.** Formally, an $n$-dimensional Lorentz model is the Riemannian manifold $\mathbb{L}_K^n = (\mathcal{L}^n, \mathfrak{g}_{\mathbf{x}}^K)$. $K$ is the constant negative curvature. $\mathfrak{g}_{\mathbf{x}}^K = diag(-1, 1, \cdots, 1)$ is the Riemannian metric tensor. We denote $\mathcal{L}^n$ as the n-dimensional hyperboloid manifold with constant negative curvature $K$:

$$\mathcal{L}^n := \left\{ \mathbf{x} \in \mathbb{R}^{n+1} : \langle \mathbf{x}, \mathbf{x} \rangle_{\mathcal{L}} = \frac{1}{K}, x_0 > 0 \right\}. \tag{1}$$

Let $\mathbf{x}, \mathbf{y} \in \mathbb{R}^{n+1}$, then the Lorentzian scalar product is defined as:

$$\langle \mathbf{x}, \mathbf{y} \rangle_{\mathcal{L}} := -x_0 y_0 + \sum_{i=1}^n x_i y_i, \tag{2}$$

where $\mathcal{L}^n$ is the upper sheet of hyperboloid in an (n+1)-dimensional Minkowski space with the origin $\left( \sqrt{-1/K}, 0, \cdots, 0 \right)$. For simplicity, we denote point $x$ in the Lorentz model as $x \in \mathbb{L}_K^n$.

**Tangent Space.** Given the tangent space at $x$ is defined as an n-dimensional vector space approximating $\mathbb{L}_K^n$ around $x$,

$$\mathcal{T}_{\mathbf{x}} \mathbb{L}_K^n := \left\{ \mathbf{y} \in \mathbb{R}^{n+1} \mid \langle \mathbf{y}, \mathbf{x} \rangle_{\mathcal{L}} = 0 \right\}. \tag{3}$$

Note that $\mathcal{T}_{\mathbf{x}} \mathbb{L}_K^n$ is a Euclidean subspace of $\mathbb{R}^{n+1}$. Particularly, we denote the tangent space at the origin as $\mathcal{T}_{\mathbf{0}} \mathbb{L}_K^n$.

**Logarithmic and Exponential Maps.** The mapping between hyperbolic spaces and tangent spaces can be done by exponential map and logarithmic map. The exponential map is a map from a subset of a tangent space of $\mathbb{L}_K^n$ (i.e., $\mathcal{T}_{\mathbf{x}} \mathbb{L}_K^n$) to $\mathbb{L}_K^n$ itself. The logarithmic map is the reverse map that maps back to the tangent space. For points $\mathbf{x}, \mathbf{y} \in \mathbb{L}_K^n$, $\mathbf{v} \in \mathcal{T}_{\mathbf{x}} \mathbb{L}_K^n$, such that $\mathbf{v} \neq \mathbf{0}$ and $\mathbf{x} \neq \mathbf{y}$, the exponential map $\exp_{\mathbf{x}}^K(\cdot)$ and logarithmic map $\log_{\mathbf{x}}^K(\cdot)$ are given as follows:

$$\exp_{\mathbf{x}}^K(\mathbf{v}) = \cosh(\sqrt{-K} \|\mathbf{v}\|_{\mathcal{L}}) \mathbf{x} + \sinh(\sqrt{-K} \|\mathbf{v}\|_{\mathcal{L}}) \frac{\mathbf{v}}{\sqrt{-K} \|\mathbf{v}\|_{\mathcal{L}}} \tag{4}$$

$$\log_{\mathbf{x}}^K(\mathbf{y}) = d_{\mathbb{L}}^K(\mathbf{x}, \mathbf{y}) \frac{\mathbf{y} - K \langle \mathbf{x}, \mathbf{y} \rangle_{\mathcal{L}}}{\|\mathbf{y} - K \langle \mathbf{x}, \mathbf{y} \rangle_{\mathcal{L}}\|_{\mathcal{L}}}, \tag{5}$$

where $\|\mathbf{v}\|_{\mathcal{L}} = \sqrt{\langle \mathbf{v}, \mathbf{v} \rangle_{\mathcal{L}}}$ denotes Lorentzian norm of $\mathbf{v}$ and $d_{\mathcal{L}}^K(\cdot, \cdot)$ denotes the Lorentzian intrinsic distance function between two points $\mathbf{x}, \mathbf{y} \in \mathbb{L}_K^n$, which is given as:

$$d_{\mathcal{L}}^K(\mathbf{x}, \mathbf{y}) = arcosh(K \langle \mathbf{x}, \mathbf{y} \rangle_{\mathcal{L}}). \tag{6}$$

# 4   Methodology

In this paper, we propose the Dual-Space Representation Learning (DSRL) method to improve the discrimination of ambiguous violence. Within DSRL, we first design the Hyperbolic Energy-constrained Graph Convolutional Network Module (HE-GCN) to better capture the hierarchical context of events (Sec. 4.1 ). Then, the Dual-Space Interaction Module (DSI) is introduced to break the information cocoon for better dual-space cooperation (Sec. 4.2 ). An illustration of the main components of DSRL is provided in Figure 2.

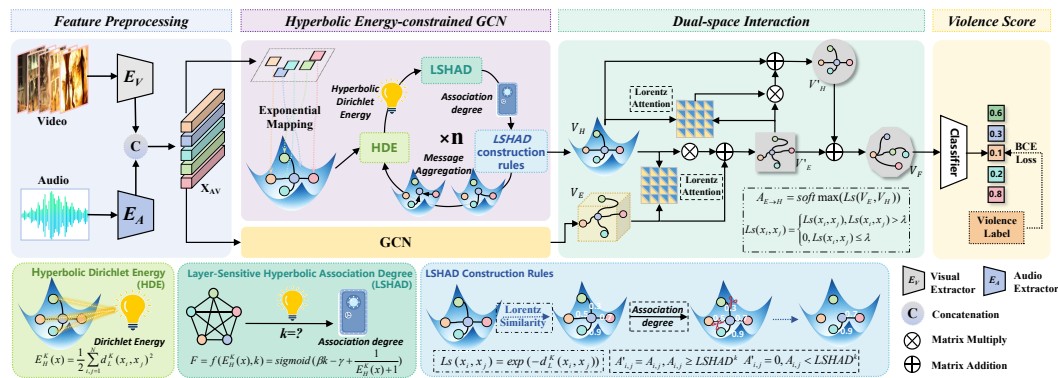

Figure 2: A conceptual diagram of our DSRL.

## 4.1 Hyperbolic Energy-constrained Graph Convolutional Network Module (HE-GCN)

HE-GCN primarily involves mapping features from Euclidean space to hyperbolic space, then transforming the features, constructing the message graph by calculating hyperbolic Dirichlet energy and Layer-Sensitive Hyperbolic Association Degree, and finally aggregating messages to obtain the new feature graph.

**Mapping from Euclidean to hyperbolic spaces.** Let $\left\{x_i^E\right\}_{i\in\mathcal{V}}$ be input Euclidean node features, and $\mathbf{o} := [1, 0, \cdots, 0]$ denote the origin on the manifold $\mathcal{L}$ of the Lorenzt model. There is $\left\langle \mathbf{o}, \left[0, x_i^E\right]\right\rangle_{\mathcal{L}} = 0$, where $\langle\cdot,\cdot\rangle_{\mathcal{L}}$ denotes the Lorentz inner product defined in Eq. 2. We can reasonably regard $\left[0, x_i^E\right]$ as a node on the tangent space at the origin $\mathbf{o}$. HE-GCN uses the exponential map defined in Eq. 4 to generate hyperbolic node representations on the Lorentz model:

$$x_i^{\mathcal{L}} = \exp_{\mathbf{o}}\left(\left[0, x_i^E\right]\right) = \left[cosh\left(\left\|x_i^E\right\|_2\right), sinh\left(\left\|x_i^E\right\|_2\right)\frac{x_i^E}{\left\|x_i^E\right\|_2}\right]. \tag{7}$$

**Hyperbolic Feature Transformation.** According to [5], we reformalize the lorentz linear layer to learn a matrix $\mathbf{M} = \begin{bmatrix}\mathbf{v}^\top \\ \mathbf{W}\end{bmatrix}$, $\mathbf{v} \in \mathbb{R}^{n+1}$, $\mathbf{W} \in \mathbb{R}^{m\times(n+1)}$ satisfying $\forall\mathbf{x} \in \mathbb{L}^n$, $f_{\mathbf{x}}(\mathbf{M})\mathbf{x} \in \mathbb{L}^m$, where $f_{\mathbf{x}} : \mathbb{R}^{(m+1)\times(n+1)} \to \mathbb{R}^{(m+1)\times(n+1)}$ should be a function that maps any matrix to a suitable one for the hyperbolic linear layer. Specifically, $\forall\mathbf{x} \in \mathbb{L}_K^n$, $\mathbf{M} \in \mathbb{R}^{(m+1)\times(n+1)}$, $f_{\mathbf{x}}(\mathbf{M})$ is given as

$$f_{\mathbf{x}}(\mathbf{M}) = f_{\mathbf{x}}\left(\begin{bmatrix}\mathbf{v}^\top \\ \mathbf{W}\end{bmatrix}\right) = \begin{bmatrix}\frac{\sqrt{\|W\mathbf{x}\|^2 - 1/K}}{\mathbf{v}^\top\mathbf{x}}\mathbf{v}^\top \\ \mathbf{W}.\end{bmatrix} \tag{8}$$

**Theorem 1** $\forall\boldsymbol{x} \in \mathbb{L}^n$, $\boldsymbol{M} \in \mathbb{R}^{(m+1)\times(n+1)}$, we have $f_x(\boldsymbol{M})\boldsymbol{x} \in \mathbb{L}_K^m$.

For simplicity, we use a morel general formula $*$ of hyperbolic linear layer for feature transformation based on $f_{\mathbf{x}}\left(\begin{bmatrix}\mathbf{v}^\top \\ \mathbf{W}\end{bmatrix}\right)\mathbf{x}$ with activation, dropout, bias and normalization,

$$\mathbf{y} = HL(\mathbf{x}) = \begin{bmatrix}\sqrt{\|\phi(\mathbf{W}\mathbf{x}, \mathbf{v})\|^2 - 1/K} \\ \phi(\mathbf{W}\mathbf{x}, \mathbf{v}),\end{bmatrix} \tag{9}$$

where $\mathbf{x} \in \mathbb{L}_K^n$, $\mathbf{W} \in \mathbb{R}^{m\times(n+1)}$, $\mathbf{v} \in \mathbb{R}^{n+1}$ denotes a velocity (ratio to the speed of light) in the Lorentz transformations, and $\phi$ is an operation function: for the dropout, the function is $\phi(\mathbf{W}\mathbf{x}, \mathbf{v}) = \mathbf{W}dropout(\mathbf{x})$; for the activation and normalization $\phi(\mathbf{W}\mathbf{x}, \mathbf{v}) = \frac{\lambda\sigma\left(\mathbf{v}^\top\mathbf{x}+b'\right)}{\|\mathbf{W}h(\mathbf{x})+\mathbf{b}\|}(\mathbf{W}h(\mathbf{x}) + \mathbf{b})$, where $\sigma$ is the sigmoid function, $\mathbf{b}$ and $b'$ are bias terms, $\lambda > 0$ controls the scaling range, $h$ is the activation function. And then, we need to construct the message graph.

**Hyperbolic Dirichlet Energy.** Given the hyperbolic embeddings $\mathbf{x} = \left\{\mathbf{x}_i \in \mathbb{L}_K^d\right\}_{i=1}^{|\mathcal{V}|}$, the hyperbolic Dirichlet energy (*HDE*) $E_H^K(\mathbf{x})$ is defined as

$$E_H^K(\mathbf{x}) = \frac{1}{2}\sum_{i,j=1}^N d_{\mathcal{L}}^K\left(\exp_{\mathbf{o}}^K\frac{log_{\mathbf{o}}^K(\mathbf{x}_i)}{\sqrt{1+d_i}}, \exp_{\mathbf{o}}^K\frac{log_{\mathbf{o}}^K(\mathbf{x}_j)}{\sqrt{1+d_j}}\right)^2, \tag{10}$$

where $d_{i/j}$ denotes the node degree of node $i/j$. The distance $d_{\mathcal{L}}^K(\mathbf{x}, \mathbf{y})$ between two points $\mathbf{x}, \mathbf{y} \in \mathbb{L}$ is the geodesic. Given that each node is connected to every other node in videos, resulting in a node degree $d_i$ of $n-1$ (where $n$ is the total number of nodes), the formula for hyperbolic Dirichlet energy can be simplified.

$$E_H^K(\mathbf{x}) = \frac{1}{2} \sum_{i,j=1}^N d_{\mathcal{L}}^K(\mathbf{x}_i, \mathbf{x}_j)^2. \tag{11}$$

*HDE* is used to measure the similarity between node features in order to gauge the degree of information aggregation among features. It is evident that as hyperbolic message aggregation progresses, the similarity between features gradually decreases. Hyperbolic message aggregation reduces *HDE*. It can be expressed as: $E_H^K(\mathbf{x}^{(l+1)}) \leq E_H^K(\mathbf{x}^{(l)})$, where $l$ is the layer number.

**Layer-Sensitive Hyperbolic Association Degree.** Based on *HDE*, we design *Layer-Sensitive Hyperbolic Association Degree (LSHAD)* to guide the node selection strategy for our construction of graphs for message aggregation. It is defined as

$$LSHAD_k = f\left(E_H^K(x), k\right) = sigmoid(\beta k - \gamma + \frac{1}{E_H^K(x) + 1}), \tag{12}$$

where $f(\cdot)$ is a function related to $k$ and $E_H^K(x)$, $k$ is the current layer number. $\beta$ and $\gamma$ are hyperparameters.

**Lorentzian similarity.** Based on the Lorentzian distance, the Lorentzian similarity to measure the feature semantic similarity between nodes is given by

$$Ls(x_i, x_j) = exp(-d_{\mathcal{L}}^K(x_i, x_j)), \tag{13}$$

where $d_{\mathcal{L}}^K(\cdot, \cdot)$ is the Lorentzian intrinsic distance function. We define the initial adjacent matrix $A^{\mathbb{L}} \in \mathbb{R}^{T \times T}$ via lorentz similarity:

$$A_{i,j}^{\mathbb{L}} = softmax(Ls(x_i, x_j)). \tag{14}$$

*LSHAD* **Construct rules.** With *LSHAD*, we propose message graph construction rules, called *LSHAD* construct rules. It is defined as

$$\begin{cases} A_{i,j}' = A_{i,j}, & \text{if } A_{i,j} \geq LSHAD^k \\ A_{i,j}' = 0, & \text{if } A_{i,j} < LSHAD^k, \end{cases} \tag{15}$$

where $A_{i,j}$ means the lorentz similarity between nodes $i$ and $j$ in the graph $G$. Formally, it enforces the elements of the adjacency matrix A that are less than *LSHAD* to be zeros. Finally, we can dynamically construct message graphs at each layer via *LSHAD* construct rules to obtain better contextual information. Following *LSHAD* construction rules, we can dynamically construct message graphs $G'$, which we then use to perform message aggregation operations to obtain better contextual information.

**Hyperbolic Message Aggregation.** We use graph $G'$ to perform message aggregation, and the message aggregation can be defined as:

$$MA(\mathbf{y}_i) = \frac{\sum_{j=1}^m A_{ij}\mathbf{y}_j}{\sqrt{-K}\left|\|\sum_{k=1}^m A_{ik}\mathbf{y}_k\|_{\mathcal{L}}\right|}, \tag{16}$$

where $m$ is the number of nodes. $\mathbf{y}_i$ is the node features.

## 4.2 Dual-Space Interaction Module

Although hyperbolic representation learning enhances understanding of event hierarchies, visual representations remain crucial in violence detection. Fusing representations from different spaces is challenging; thus, DSI employs cross-space attention to facilitate interactions between Euclidean and hyperbolic spaces.

**Cross-Space Attention Mechanism.** Cross-Space Attention Mechanism utilizes the Lorentzian metric to calculate attention scores between nodes from different spaces, accurately measuring semantic similarity and better preserving their true relationships by computing the nonlinear distance between them. We denote the features in Euclidean space as $V_E$ and the features in hyperbolic space as $V_H$. $CSA_{E \rightarrow H}$ models the between-graph interaction and guides the transfer of inter-graph

message from $V_E$ to $V_H$. First, we use linear layer to transform $V_H$ to the *key* graph $V_k$ and the *value* graph $V_v$, and $V_E$ to the *query* graph $V_q$. Then, we use lorentzian metric to calculate the attention map $\mathcal{A}_{E \rightarrow H}$ as follows:

$$\mathcal{A}_{E \rightarrow H} = softmax(Ls(V_q, V_k)), Ls(x_i, x_j) = \begin{cases} Ls(x_i, x_j), & \text{if } Ls(x_i, x_j) > \lambda \\ 0, & \text{if } Ls(x_i, x_j) \leq \lambda, \end{cases} \tag{17}$$

where $Ls(\cdot)$ is Lorentzian similarity defined in Eq. 13 and $\lambda$ is the threshold value to eliminate weak relations and strengthen correlations of more similar pairs. The representation from $E$ to $H$ can be formulated as follows:

$$V_H' = CSA_{E \rightarrow H}(V_H, V_E) = softmax(\frac{\mathcal{A}_{E \rightarrow H} \times V_k}{\sqrt{d}})V_v. \tag{18}$$

The interaction process in DSI can be represented as follows:

$$\begin{aligned} V_E' &= \alpha \times CSA_{E \rightarrow H}(V_E, V_H) + V_E \\ V_H' &= \alpha \times CSA_{H \rightarrow E}(V_H, V_E') + V_H \\ V_F &= MaxPool([V_E' \oplus V_H']), \end{aligned} \tag{19}$$

where $V_E'$ represents the features obtained by enhancing Euclidean space features using hyperbolic space features, and $\alpha$ controls the contribution of the enhanced features and $V_E$. $MaxPool$ is the max pooling operation and $\oplus$ represents the concatenation operation.

**Learning Objective.** We use binary cross-entropy as our classification loss. Its calculation formula is:

$$Loss = -\frac{1}{N} \sum_{i=1}^{N} (y_i \log(\hat{y}_i) + (1 - y_i)log(1 - \hat{y}_i)), \tag{20}$$

where $y_i$ is true label, $\hat{y}_i$ is the predicted label, $N$ is the batch size.

# 5 Experiments

## 5.1 Experiments Setup

**Datasets.** Under the multimodal input setting, we follow [34, 37, 25] to conduct experiments on XD-Violence, which is the only and extremely challenging VVD dataset with multimodal information. Under the unimodal input setting, both the XD-Violence[34] and UCF-Crime[30] datasets are used to evaluate our method. More details of the two dataset settings are provided in the Appendix.

**Evaluation Metrics.** To quantitatively evaluate the performance, we follow standard pratice [34, 37, 18, 6]. For XD-Violence, we utilize the frame-level average precision (AP) as the evaluation metric. For UCF-Crime, we adopt the area under the curve of the frame-level receiver operating characteristic (AUC) to evaluate performance.

## 5.2 Comparisons with State-of-the-art Methods

In Table 1, with multimodal input, our DSRL demonstrates superior performance on XD-Violence, surpassing the best method that uses Euclidean space representation by 4.21% and that uses only hyperbolic space representation by 1.94%. These results highlight the effectiveness of DSRL in learning discriminative features and prove that our dual-space representation learning integrates the benefits of Euclidean and hyperbolic space well. From Table 1, it also can be found that DSRL achieves SOTA performance under unimodal input settings on XD-Violence, outperforming existing state-of-the-art methods. Furthermore, to analyse the generalization of our method, we also report the performance on the UCF-Crime dataset, achieving an accuracy of 86.38%, comparable to current state-of-the-art methods.

## 5.3 Ablation Studies

We conduct ablation studies on various design choices of our DSRL to demonstrate their contributions to the final results in Table 2.

Table 1: Comparisons of frame-level AP performance on XD-Violence and AUC performance on UCF-Crime datasets under different input settings. UCF-Crime only has visual modality input.

| Methods | Input Setting | Feature Space | UCF-Crime | XD-Violence |
|---|---|---|---|---|
| Sultani et al. [30] | Unimodal | Euclidean | 76.21 | 73.20 |
| Wu et al. [33] | Unimodal | Euclidean | 82.44 | 75.90 |
| RTFM [31] | Unimodal | Euclidean | 84.30 | 77.81 |
| MSL [16] | Unimodal | Euclidean | 85.30 | 78.28 |
| MGFN [6] | Unimodal | Euclidean | **86.98**($1^{st}$) | 79.19($3^{rd}$) |
| UMIL [18] | Unimodal | Euclidean | 86.75($2^{nd}$) | 81.66($2^{nd}$) |
| CU-Net [39] | Unimodal | Euclidean | 86.22 | 78.74 |
| **Ours** | Unimodal | Euclidean and Hyperbolic | 86.38($3^{rd}$) | **82.01**($1^{st}$) |
| HL-Net [34] | Multimodal | Euclidean | - | 78.64 |
| Wu et al. [35] | Multimodal | Euclidean | - | 78.64 |
| Pang et al. [23] | Multimodal | Euclidean | - | 79.37 |
| UMIL [18] | Multimodal | Euclidean | - | 81.77 |
| Zhang et al. [39] | Multimodal | Euclidean | - | 81.43 |
| MACIL-SD [37] | Multimodal | Euclidean | - | 83.40($3^{rd}$) |
| HyperVD [25] | Multimodal | Hyperbolic | - | 85.67($2^{nd}$) |
| **Ours** | Multimodal | Euclidean and Hyperbolic | - | **87.61**($1^{st}$) |

Table 2: Ablations on XD-Violence dataset.

| Euclidean | Hyperbolic | | DSI | | | XD-Violence | |
|---|---|---|---|---|---|---|---|
| GCN | HE-GCN | HGCN | Concat | Cosine Metric | Lorentzian Metric | Multimodal(%) | Unimodal(%) |
| ✓ | | | | | | 84.04 | 77.95 |
| ✓ | | ✓ | ✓ | | | 85.01 | 77.93 |
| ✓ | ✓ | | ✓ | | | 86.46 | 79.70 |
| ✓ | ✓ | | | ✓ | | 86.91 | 80.72 |
| ✓ | ✓ | | | | ✓ | 87.61 | 82.01 |

1) **Component-wise ablations.** To study the impact of each component in the DSRL, including HE-GCN and DSI, we start with a baseline model that only applies GCN and progressively adds each component. Table 2 shows that the baseline yields inferior performance due to learning representations only in Euclidean space ($1^{st}$ row). Then, HE-GCN is employed and the representations from different spaces are simply concat at this stage, benefit from the hierarchical context of events, resulting in an improvement of 2.42% on multimodal setting and 1.75% on unimodal setting respectively ($3^{rd}$ row). Then DSI is introduced to facilitate information interactions, which further enhances the performance on multimodal input setting by 1.15% and unimodal input setting by 2.31% ($5^{th}$ row). 2) **Effect of *LSHAD*.** *LSHAD* is a crucial element of our HE-GCN for constructing the message graph. In Table 2($2^{nd}$ and $3^{rd}$ rows), we present experiments comparing HGCN used in HyperVD [25], which employs a hard node selection strategy (nodes selected by a fixed threshold), with HE-GCN, which uses our introduced layer-sensitive hyperbolic association degrees for node selection in message aggregation. The performance of AP improved by 1.45% in the multimodal setting and by 1.77% on the unimodal setting. These results demonstrate that our *LSHAD* is a better strategy and helps to capture the hierarchical context of events. 3) **Lorentzian metric in DSI.** As shown in Table 2($4^{th}$ and $5^{th}$ rows), we also explore the effect of cosine similarity and Lorentzian metric in our DSI, and the results show that using cosine similarity to calculate the attention between nodes resulted in a 0.7% lower performance compared with using the Lorentzian metric. This indicates that the Lorentzian metric is more effective in measuring feature similarity across different spaces.

## 5.4 Qualitative Results

1) **Feature Discrimination Visualization.** We present t-SNE visualizations of feature distributions on the XD-Violence test set. As shown in Figure 3, red dots represent violent features, while purple dots represent normal features. The clear clustering of violent and non-violent features demonstrates the effectiveness of DSRL.

2) **Qualitative Visualizations.** We illustrate the qualitative visualizations of VVD for the test video

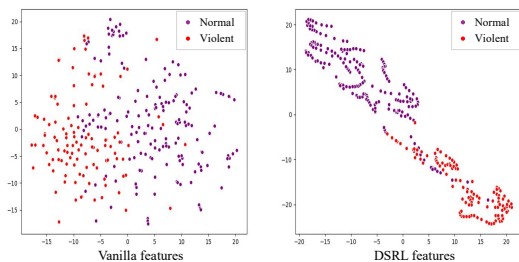

Figure 3: t-SNE visualization of vanilla and DSRL features for the test video on XD-Violence.

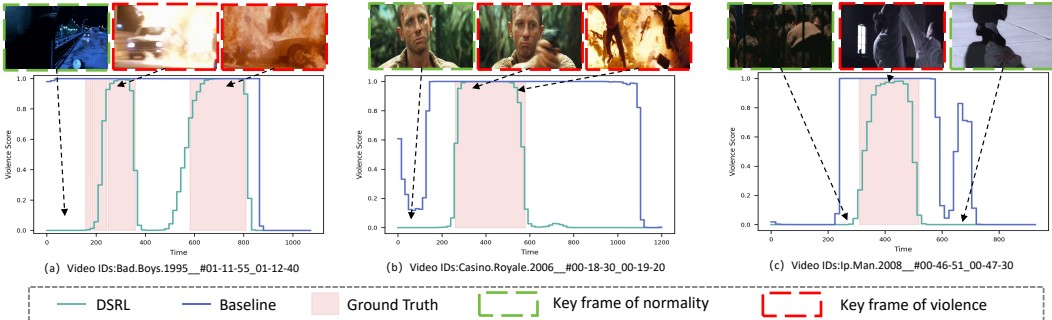

Figure 4: Frame-level scores and violence localization examples for the test video from XD-Violence dataset.

from the XD-Violence dataset. Figure 4 shows that DSRL can accurately detect violent events and has a better detection performance than the baseline.

3) **Qualitative Visualizations of DSRL in the Context of Ambiguous Violence.** We put up qualitative visualizations of DSRL when handling ambiguous violence. Figure 5 demonstrates that DSRL effectively resolves ambiguous violence, which single-space representation learning struggles with. In Figure 5(a), the first frame shows smoke caused by fire. Euclidean space representation, relying on visual features, misidentifies the smoke as violence. Hyperbolic space representation, considering contextual information, also misidentifies it due to preceding violent frames. DSRL, however, combines both perspectives: Euclidean space flags the smoke as a potential violence indicator, while hyperbolic space recognizes the fire context. This integration allows DSRL to accurately classify the smoke as non-violent. This supports our motivation: hyperbolic representation enhances hierarchical event relations but weakens visual feature expression, while Euclidean representation emphasizes visual features but overlooks event relationships. DSRL effectively addresses ambiguous violence, which is challenging for either space alone. More performance visualizations are included in the Appendix.

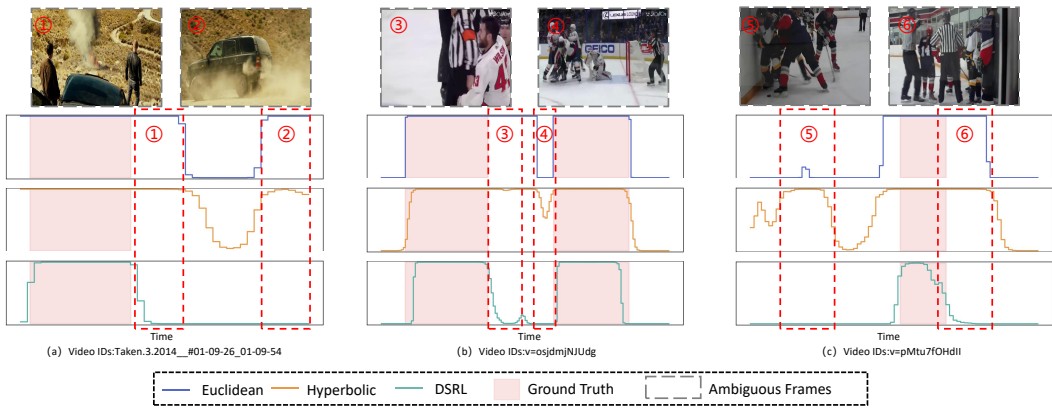

Figure 5: Some visual results of DSRL in the context of ambiguous violence. "Euclidean" represents the results of GCN only. "Hyperbolic" refers to the results of HyperVD.

## 5.5 Comparisons of Computing Resources and Training Time

Our designs, HE-GCN and DSI, are specifically crafted to effectively capture hierarchical contextual information of events and to integrate two distinct spaces of information, respectively. To evaluate the benefits of these designs, we conducted experiments on the XD-Violence dataset, training three models for 30 epochs each on a single NVIDIA RTX A6000 GPU: the Baseline (GCN), Baseline+HE-GCN, and Baseline+HE-GCN+DSI (referred to as DSRL). As shown in Table 3, our DSRL model demonstrated a 3.57% improvement in Average Precision (AP) compared to the Baseline. Additionally, the training time per epoch increased by only 41 seconds, and memory usage rose by just 4.1 GB. Both increases are within a reasonable range, making the performance gains achieved by DSRL well worth the additional resource consumption.

Table 3: Comparison of computing resources and training time.

| Methods | Params | Training time per epoch | Training time | Video memory usage | AP (%) |
|---|---|---|---|---|---|
| Baseline(GCN) | 0.7734M | 2min | 60min | 4.24GB | 84.04 |
| Baseline+HE-GCN | 0.8975M | 2min19s | 69min39s | 7.03GB | 86.46 |
| Baseline+HE-GCN+DSI(DSRL) | 0.9966M | 2min41s | 80min19s | 8.34GB | 87.61 |

## 5.6 Analysis of Model Computational Complexity and Speed

The computational efficiency of the DSRL model is crucial, especially for real-time applications. Our analysis confirms that the model meets the requirements for real-time processing, as demonstrated by the following results. Our experiments were conducted on a single NVIDIA RTX A6000 GPU. For **video input**, the model processes at a rate of 83.87 FPS, handling only video data. The model's parameters are relatively lightweight, totaling 13.4 MB, with I3D parameters at 12.49 MB and DSRL parameters at 0.91 MB. This compact size ensures quick response times. When processing **video and audio inputs**, the model maintains a high processing speed of 56.86 FPS, despite the added computational load from audio processing. The total parameter size for this configuration is 85.54 MB, comprising I3D parameters at 12.49 MB, VGGish parameters at 72.14 MB, and DSRL parameters at 0.91 MB. These results illustrate that our model exhibits excellent real-time performance, even with multimodal inputs, making it suitable for latency-sensitive real-world applications.

# 6 Conclusions

In this paper, we propose a comprehensive geometric representation learning method, Dual-Space Representation Learning (DSRL) which integrates the benefits of Euclidean and hyperbolic geometries to improve the discrimination of ambiguous violence. Hyperbolic Energy-constrained Graph Convolutional Network (HE-GCN) is designed to better capture the hierarchical context of events. Additionally, Dual-Space Interaction (DSI) is designed to facilitate information interactions. Our method achieves SOTA performance on the XD-Violence dataset in both unimodal and multimodal settings, especially excelling in resolving ambiguous violence.

**Limitations.** Our DSRL is effective for VVD in a multimodal input setting. However, DSRL only utilizes basic audio information, potentially overlooking the more detailed semantic content present in the audio. How to further narrow this limitation is our future research focus. Due to the bias issues in the XD-Violence and UCF-Crime datasets, which fail to adequately represent diverse backgrounds, the fairness and generalization of the model may be affected. In the future, we hope to use more diverse datasets and conduct bias analysis to prevent the model from producing unfair outcomes due to false associations with gender or race.

**Acknowledgements.** This work was supported in part by the Science and Technology Innovation Key R&D Program of Chongqing under Grant No. CSTB2023TIAD-STX0016, in part by the National Natural Science Foundation of China under Grants No. 62472060, 62441601, U23A20318, and 62221005, in part by the Natural Science Foundation of Chongqing under Grand No. CSTB2022NSCQ-MSX1024, CSTB2023NSCQ-LZX0061, in part by the Science and Technology Research Program of Chongqing Municipal Education Commission under Grant No. KJZD-K202300604, and in part by the Chongqing Institute for Brain and Intelligence.

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

## Appendix

Here we provide the proof of Theorem 1 in Sec. A. The Technical details of the DSRL are introduced in Sec. B. Moreover, the datasets settings and implementation details are shown in Sec. C. We conduct additional experiments in Sec. D, and more qualitative results in Sec. E. Broader impacts are listed in Sec.F.

## A    Proof of Theorem. 1

**Theorem 1.** $\forall \mathbf{x} \in \mathbb{L}^n$, $\mathbf{M} \in \mathbb{R}^{(m+1) \times (n+1)}$, we have $f_x(\mathbf{M})\mathbf{x} \in \mathbb{L}_K^m$.

*Proof 1.*

$$\mathbf{y} = f_{\mathbf{x}}(\mathbf{M})\mathbf{x} = \begin{bmatrix} \frac{\sqrt{\|W\mathbf{x}\|^2 - 1/K}}{\mathbf{v}^\top \mathbf{x}} \mathbf{v}^\top \mathbf{x} \\ \mathbf{W}\mathbf{x} \end{bmatrix} = \begin{bmatrix} \sqrt{\|W\mathbf{x}\|^2 - 1/K} \\ \mathbf{W}\mathbf{x} \end{bmatrix}$$

$$\langle \mathbf{y}, \mathbf{y} \rangle_{\mathcal{L}} = -(\|W\mathbf{x}\|^2 - 1/K) + (\mathbf{W}\mathbf{x})^\top \mathbf{W}\mathbf{x}$$
$$= \frac{1}{K} - \|W\mathbf{x}\|^2 + \|W\mathbf{x}\|^2$$
$$= \frac{1}{K}$$

Thus, $f_x(\mathbf{M})\mathbf{x}$ lies on the manifold $\mathcal{L}$ of the Lorentz model.

## B    Technical details of the DSRL

As depicted in Figure 2, the entire process can be divided into feature preprocessing to integrate the features of the two modalities; representation learning in hyperbolic space to learn the hierarchical context of events; representation learning in Euclidean space to learn the visual features of events; interaction between the two spaces to promote cross-space enhancement; and finally, feeding the features into a hyperbolic classifier for classification.

**Feature Preprocessing.** Following the previous works[34, 23], the visual and audio segments are processed by the I3D [2] network pretrained on the Kinetics-400 dataset and the VGGish[11] network pretrained on a large YouTube dataset, respectively. After that, we perform further feature extraction using simple convolution and pooling operations. A simple cross-modal attention mechanism is then employed to enhance the audio features, which are subsequently concatenated with the visual features to form the fused features.

**Hyperbolic Representation Learning.** In this part, we first use HE-GCN to learn the hierarchical context of events. Meanwhile, the temporal relation is also crucial for numerous video-based tasks. Therefore, we construct a temporal relation graph directly based on the temporal structure of a video and learn the temporal relation among snippets in hyperbolic space via HGCN. Its adjacency matrix $A^{\mathbb{T}} \in \mathbb{R}^{T \times T}$ is only dependent on temporal positions of the $i$-th and $j$-th snippets, which can be defined as:

$$A_{ij}^{\mathbb{T}} = exp(-\frac{|i-j|}{\sigma}) \tag{21}$$

where $\sigma$ controls the range of influence of distance relation. Finally, we use HE-GCN for semantic message aggregation and HGCN for temporal message aggregation, then concatenate them to obtain a hyperbolic space representation.

**Euclidean Representation learning.** The process of representation learning in Euclidean space is similar to that in hyperbolic space and follows a dual-branch structure, considering both semantic and temporal relationships. We use cosine similarity to compute the semantic similarity in Euclidean space and use Eq. 21 to compute temporal relationships. Finally, we employ GCN for message aggregation and concatenate them to obtain the final representation in Euclidean space.

**Dual-Space Interaction.** In this part, we primarily use cross-space attention to enhance the interaction between the features of the two spaces. The detailed process is described in Sec. 4.2.

**Hyperbolic Classifier.** As shown in Figure.2, we input the enhanced embeddings from DSI into hyperbolic classifier utilizing Lorentzian metric, which can be formalized as:

$$S = \sigma(\epsilon + \epsilon \langle F, W \rangle_{\mathcal{L}} + b) \tag{22}$$

where $\sigma$ is sigmoid function and $W$ is weight matrices. $b$ and $\epsilon$ denotes bias term and hyper-parameter, respectively. Lastly, we supervise the training of the violence scores obtained by the model with the real labels using the binary cross-entropy loss.

## C   Experimental Details

**Dataset.** We conducted experiments on XD-Violence with both multi-modal input settings and single-modal input settings. Additionally, we performed experiments on UCF-Crime with single-modal input settings to demonstrate the generalization capability of DSRL.
1) **XD-Violence** dataset is by far the only available large-scale audio-visual dataset for violence detection, which is also the largest dataset compared with other unimodal datasets. XD-Violence consists of 4,757 untrimmed videos (217 hours) and six types of violent events, which are curated from real-life movies and in-the-wild scenes on YouTube. For XD-Violence dataset, only video-level annotations are provided.
2) **UCF-Crime** dataset is a large-scale dataset comprised of real-world videos captured by surveillance cameras. It consists of 1,610 training videos annotated with video-level labels and 290 test videos annotated at the frame level to facilitate performance evaluation. The videos are collected from different scenes and encompass 13 distinct categories of anomalies.
**Implementation Details.** The visual sample rate is set to 24 fps, and visual features are extracted by a sliding window with a size of 16 frames. For the audio data, we first divide each audio into 960-ms overlapped segments and compute the log-mel spectrogram with $96 \times 64$ bins. Our proposed method is trained for 30 epochs in total, and the batch size is 256. The initial learning rate is 0.001, which is dynamically adjusted by a cosine annealing scheduler [13]. We use Adam [14] as the optimizer without weight decay. For hyper-parameters, we set $\beta$ as 0.8, $\gamma$ as 1.2, $\alpha$ as 0.3, and dropout rate as 0.6. Following [25], $\sigma$ is empirically set to $e$. For the MIL, we set the value $k$ of $k$-max activation as $\lfloor \frac{T}{16} + 1 \rfloor$, where $T$ denotes the length of the input feature.
**Experiments Compute Resources.** We use an Intel(R) Xeon(R) Platinum 8260 CPU @ 2.40GHz, a NVIDIA RTX A6000 GPU to conduct experiments. We use CUDA 12.2, Python 3.9.16, and Pytorch 1.12.1.

## D   Additional Experiments

**Reasons for the design choices in LSHAD with its multiple hyperparameters and threshold criteria.**
Inspired by the Global-first principle[4] that humans always have cognition on global first and then focus on local, we propose a novel node selection strategy, which guarantees the model captures the broader global context first with a relaxed threshold at the beginning of message aggregation and then focuses on the local context with more strict thresholds. To achieve this, we introduce the LSHAD construction rule, which calculates an LSHAD threshold based on the number of the current layer $K$ and hyperbolic Dirichlet energy of the current layer. As the $K$ increases and the hyperbolic Dirichlet energy decreases, the LSHAD threshold increases and is limited to 0 and 1 by the sigmoid function. If there is no $\beta$ and $\gamma$, the threshold in the first layer will be strict ($> 0.5$), causing the overlook of some global context information. Therefore, to make our node selection threshold conform to the Global-first principle, we introduced the two hyperparameters in LSHAD, where $\beta$ controls the influence of the number of current layer $k$ and $\gamma$ acts as a bias to fine-tune the threshold. Moreover, we conducted an ablation study to determine the optimal value of the two hyperparameters ($\beta,\gamma$), where $\beta$ ranges from [0.2,0.4,0.6,0.8,1.0] and $\gamma$ ranges from [1.0,1.2,1.4,1.6,1.8,2.0]. The results in the table below reveal that when $\gamma - \beta$ is 0.4, the performance is optimal, so we chose a pair (0.8, 1.2) from this set.
**Ablation studies on some hyperparameters in DSI.**
Moreover, we conduct ablation studies on some hyperparameters in DSI. Table 5 and 6 show the experimental results.

Table 4: Ablation studies on $\beta$ and $\gamma$.

| $\beta$ \ $\gamma$ | 1.0 | 1.2 | 1.4 | 1.6 | 1.8 | 2.0 |
|---|---|---|---|---|---|---|
| 0.2 | 85.22 | 87.12 | 86.32 | 86.60 | 86.31 | 86.86 |
| 0.4 | 87.52 | 85.22 | 87.12 | 86.32 | 86.60 | 86.31 |
| 0.6 | 87.61 | 87.52 | 85.22 | 87.12 | 86.32 | 86.60 |
| 0.8 | 87.29 | 87.61 | 87.52 | 85.22 | 87.12 | 86.32 |
| 1.0 | 86.29 | 87.29 | 87.61 | 87.52 | 85.22 | 87.12 |

Table 5: Ablation studies on $\lambda$ of DSI.

| $\lambda$ | 0.1 | 0.2 | 0.3 | 0.4 | 0.5 | 0.6 | 0.7 | 0.8 | 0.9 |
|---|---|---|---|---|---|---|---|---|---|
| AV-AP | 87.14 | 87.38 | 87.1 | 87.32 | 86.53 | 87.2 | 87.4 | 87.61 | 87.23 |

Table 6: Ablation studies on $\alpha$ of DSI.

| $\alpha$ | 0.1 | 0.2 | 0.3 | 0.4 | 0.5 | 0.6 | 0.7 | 0.8 | 0.9 |
|---|---|---|---|---|---|---|---|---|---|
| AV-AP | 87.18 | 87.16 | 87.61 | 87.21 | 87.26 | 86.91 | 87.42 | 86.59 | 87.51 |

# E  Visualization

**Inference visualizations of the ablation modules.**
In this section, we have supplemented inference visualization results for the ablation module, and conducted corresponding visualisation experiments to explore the discriminative power of the model w/ or w/o our core modules, HE-GCN and DSI. We conducted the analysis from two dimensions (the feature-level and the frame-leve), and the results are shown in Figure 6 and Figure 7.

**At the feature-level,** the results are shown in Figure 6. Compared to GCN, HE-GCN can capture the hierarchical context of events, effectively separating features. This results in a greater distance between feature clusters compared to Figure 6(a) original features and Figure 6(b) using only Euclidean representation learning. However, some challenging feature points remain difficult to distinguish. The addition of the DSI module facilitates information interactions between Euclidean and hyperbolic spaces, capturing more discriminative features to better differentiate these challenging feature points. As shown in Figure 6(d), the DSI module further enhances feature differentiation by effectively combining information from both spaces.

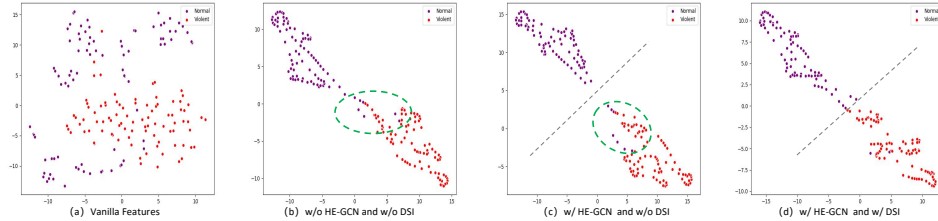

Figure 6: t-SNE visualization of the ablation module at feature-level.

Meanwhile, **at the frame-level**, experiments conducted on two test videos as shown in Figure 7 demonstrate that our method significantly improves the discriminative power for identifying violent frames compared to the baseline, which uses only GCN. Compared with the model w/o our core modules, both HE-GCN and DSI contribute to detecting violent frames.

Moreover, we provide more qualitative results, including the qualitative visualizations of VVD (Figure 8) and the qualitative visualizations of DSRL in the context of ambiguous violence (Figure 9).

**Qualitative visualizations.**
Figure 8 illustrates that DSRL can accurately distinguish between violent and normal events, demonstrating the effectiveness of DSRL. Additionally, compared with the baseline curves, our method shows better performance, further validating the effectiveness of the modules we designed.

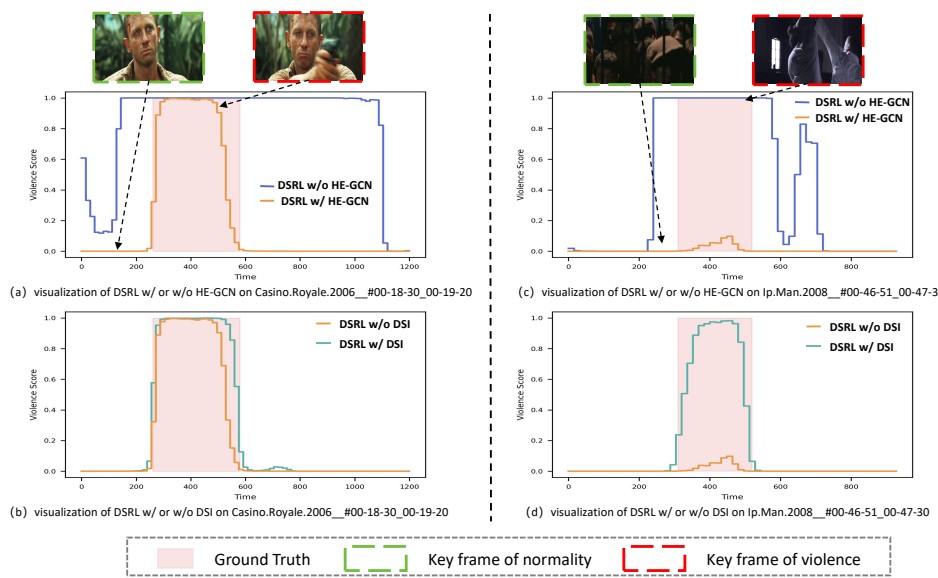

Figure 7: Qualitative results of DSRL w/ or w/o our core modules (HE-GCN and DSI) for the test video from XD-Violence dataset at frame-level.

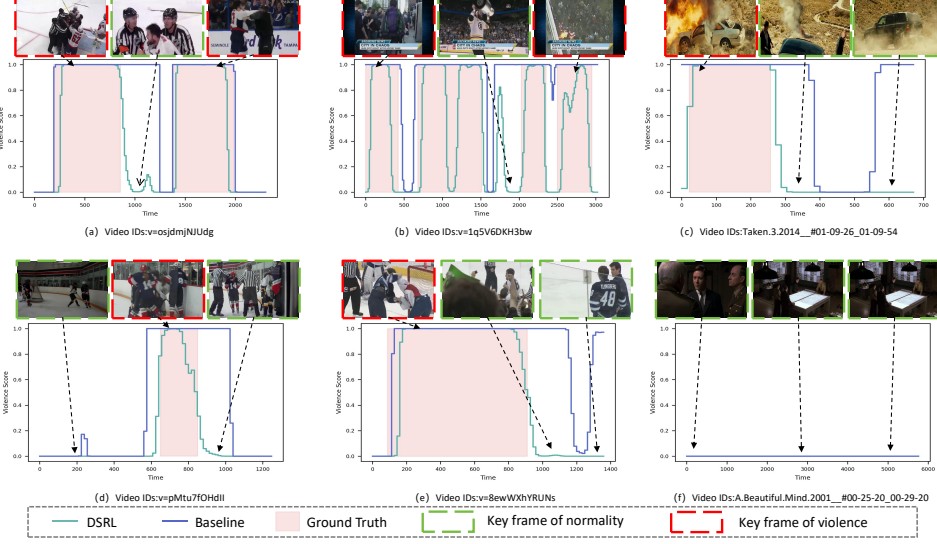

Figure 8: Frame-level scores and violence localization examples for the test video from XD-Violence dataset.

**Qualitative Visualizations of DSRL in the context of ambiguous violence.**
Figure9 shows that in the XD-Violence test set, DSRL accurately detects the correct categories of ambiguous violence, while using only Euclidean space or only hyperbolic space fails to correctly detect these instances.

# F   Broader impacts

**Potential positive societal impacts.**
Our work can be more effective in identifying incidents of violence, albeit it may be ambiguous. This contributes to a higher level of safety in the community and the public's sense of security.
**Potential negative societal impacts.**
If our algorithm were to be widely applied in monitoring and law enforcement domains, it could

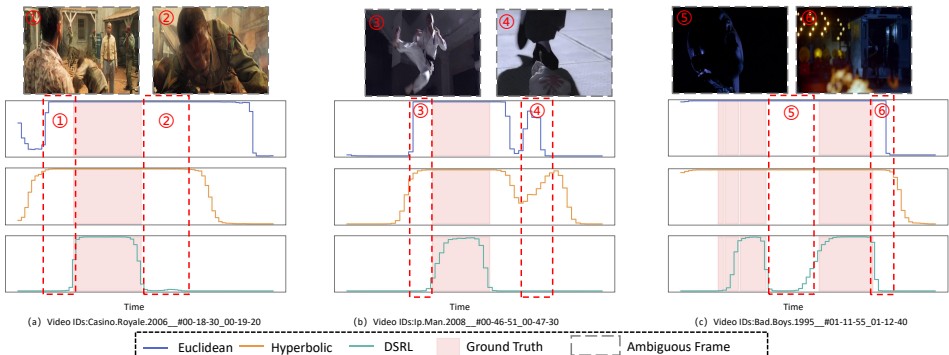

Figure 9: Qualitative Visualizations of DSRL in the context of ambiguous violence. The blue curves show violence scores predicted using only Euclidean representation, the yellow curve shows scores using only hyperbolic representation, the green curves show scores predicted by DSRL, and the pink area represents the ground-truth violent temporal location.

exacerbate societal surveillance and control, triggering concerns among the public regarding individual freedom and privacy rights.

