# OpenReview forum: "Beyond Euclidean: Dual-Space Representation Learning for Weakly Supervised Video Violence Detection"
_NeurIPS.cc/2024/Conference — NeurIPS 2024 poster_

### Official Review · Reviewer_8SDB · 2024-06-22

**Soundness:** 3
**Presentation:** 3
**Contribution:** 3
**Rating:** 6
**Confidence:** 4

**Summary:**

This paper presents a pioneering approach called Dual-Space Representation Learning (DSRL) for the task of weakly supervised video violence detection.  The overall framework contains the Hyperbolic Energy-constrained Graph Convolutional Network (HE-GCN) for capturing event hierarchies and the Dual-Space Interaction (DSI) module to facilitate cross-space feature integration. Experiments on XD-Violence and UCF-Crime datasets show the advantage of the proposed method.

**Strengths:**

- **Motivation**: The authors propose to combine Euclidean and hyperbolic geometries to handle the challenging scenarios of ambiguous violence, which is a problem-centric motivation.
- **Written**: The quality of this paper's presentation is good and the whole paper is well-organized.
- **Technical Correctness**: The proposed method is technically sound and has been evaluated on two datasets.

**Weaknesses:**

- **Novelty**: Some modules in this paper lack innovation, such as the cross-graph attention mechanism is commonly used in previous works.
- **Model Complexity**: This paper does not discuss the computational complexity or parameters of the DSRL model. For practical applications, especially in real-time surveillance scenarios, it's important to ensure that the model can operate efficiently with minimal latency.

**Questions:**

- **Qualitative Visualizations**: Does the 'Hyperbolic' in Figure 5 represent the result of HyperVD? If not, I wonder whether HyperVD can handle ambiguous violence.

**Limitations:**

Please see #Weaknesses and #Questions.

---

> ### Author Rebuttal · Authors · 2024-08-07
>
> We thank the reviewer for the positive evaluation of the paper. We have carefully considered your constructive and insightful comments and here are the answers to your concerns.
>
> ***Q1: Novelty: Some modules in this paper lack innovation, such as the cross-graph attention mechanism is commonly used in previous works.***
>
> Thank you for your feedback regarding the novelty of our work. Here we would like to emphasize the innovative aspects of the introduced Cross-Space Attention (CSA) mechanism and Hyperbolic Energy-constrained Graph Convolutional Network Module (HE-GCN).
> Compared with the commonly used cross-graph attention (CGA) mechanism,
> **i) the role of our CSA is different.** CSA aims to break the information cocoon of different spaces and realize the interaction of different geometric structure information across spaces while CGA focuses on the representation learning of the same geometric structure information in a single space.
>
> **ii) The interaction strategy of CSA is different.** On the one hand, since the Lorentzian metric preserves true relationships by computing the nonlinear distance between nodes, while the cosine similarity may show fake relationships, especially for ambiguous violence with similar visual cues, we adopt the Lorentzian metric instead of the cosine similarity that is commonly used in CGA to obtain attention weights for better interaction in spaces with different geometric structures. The corresponding comparison results reported in the submitted manuscript (lines 270-274) also prove the effectiveness of the Lorentzian metric. On the other hand, we use a step-by-step interaction model. Unlike the commonly used one-step interaction, the result of one-step interaction, which is dominated by the information of the current space, is interacted again with the original information in another space to achieve full cross-space information interaction.
>
> **iii) The construction of the graph for interaction is different.** Unlike the traditional strategy of node selection for message aggregation, the graph in CSA is constructed through a more effective dynamic node selection strategy where the node selection threshold during message aggregation is layer-sensitive and affected by the hyperbolic Dirichlet energy of the current layer.
>
> It is worth emphasizing that HE-GCN is also one of our main innovations. Instead of adopting the hard node selection strategy in HGCN, HE-GCN selects nodes for message aggregation by our introduced layer-sensitive hyperbolic association degrees, which are dynamic thresholds determined by the message aggregation degree at each layer. To better align with the characteristics of hyperbolic spaces, we introduce the hyperbolic Dirichlet energy to quantify the extent of message aggregation. Benefiting from the dynamic threshold, the layer-by-layer focused message passing strategy adopted by HE-GCN not only ensures the efficiency of information excavation but also improves the model's comprehensive understanding of the events, thus enhancing the model's ability to discriminate ambiguous violent events.
>
> ***Q2: Model Complexity: This paper does not discuss the computational complexity or parameters of the DSRL model. For practical applications, especially in real-time surveillance scenarios, it's important to ensure that the model can operate efficiently with minimal latency.***
>
> Thank you for the valuable feedback regarding the computational complexity and parameters of the DSRL model. We agree that understanding the model's complexity is especially important for real-time applications. Our model indeed meets the requirements for real-time processing, as analyzed below:
> Our experiments were conducted on a single NVIDIA RTX A6000 GPU.
> For **Video Input**, the model achieves 83.87 FPS while handling only video data. The model's parameters are 13.4 MB (I3D parameters at 12.49 MB and DSRL parameters at 0.91MB), which keeps the overall parameter size manageable and enables quick response times.
> For **Video + Audio Input**, the model maintains a high processing speed of 56.86 FPS, even with the additional computational load of audio processing. The model's parameters are 85.54 MB (I3D parameters at 12.49 MB, VGGish parameters at 72.14 MB and DSRL parameters at 0.91MB ).
>
> These results demonstrate that our model exhibits excellent real-time performance with multimodal inputs, making it suitable for latency-sensitive real-world applications.
>
> ***Q3: Does the "Hyperbolic" in Figure 5 represent the result of HyperVD?  If not, I wonder whether HyperVD can handle ambiguous violence.***
>
> Yes, the "Hyperbolic" in Figure 5 indeed represents the result of HyperVD.
> Our experiments indicate that HyperVD is not sufficiently effective in handling ambiguous violence due to two main limitations. First, HyperVD relies solely on hyperbolic representations, which weakens its ability to capture essential visual features. Second, it inadequately learns the hierarchical relationships of complex violent events, which further contributes to its suboptimal performance in dealing with ambiguous violence. We will clarify in the final version of the paper that "Hyperbolic" refers to the results of HyperVD.

---

> > ### Comment · Reviewer_8SDB · 2024-08-08
> >
> > Thanks for providing the response. It partially addressed my concerns. However, can you provide the real-time processing speed of HyperVD and MACIL-SD? It would be nice to see some comparisons between them.

---

> > > ### Author Response · Authors · 2024-08-08
> > >
> > > Thank you for the response.
> > >
> > > Following your suggestion, we have evaluated the real-time processing speed of HyperVD and MACIL-SD with an NVIDIA RTX A6000 GPU for a fair comparison, and the FPS of HyperVD and MACIL-SD is 98.90 and 102.99, respectively.
> > >
> > > In this work, we focus on addressing the problem of ambiguous violence. To tackle this issue, we have to introduce some additional computations, with the speed sacrificed to some extent. Despite our method not being as fast as these two methods, it still meets real-time processing requirements and is acceptable. Moreover, our AP performance on the XD-Violence dataset has improved by 4.21\% over MACIL-SD and 1.94\% over HyperVD. Notably, from the visualization comparison experiments, as shown in Figure 5 ("Hyperbolic" is HyperVD) of the submitted manuscript, our method is obviously better than HyperVD in handling ambiguous violence.
> > >
> > > We hope our responses have addressed your concerns. If you have any further questions or comments, please kindly let us know, and we will be glad to respond.

---

> > > > ### Comment · Reviewer_8SDB · 2024-08-09
> > > >
> > > > Thanks for your feedback! I will keep my original rating.

---

### Official Review · Reviewer_5VkS · 2024-07-07

**Soundness:** 3
**Presentation:** 3
**Contribution:** 3
**Rating:** 7
**Confidence:** 5

**Summary:**

This paper presents a novel approach called Dual-Space Representation Learning (DSRL) for weakly supervised Video Violence Detection (VVD). Traditional VVD methods rely heavily on Euclidean space representation, which often fails to distinguish between visually similar events. The proposed DSRL method leverages both Euclidean and hyperbolic geometries to enhance the discriminative capacity of features. It introduces two key modules: the Hyperbolic Energy-constrained Graph Convolutional Network (HE-GCN) and the Dual-Space Interaction (DSI) module, to facilitate better information interactions and improve violence detection accuracy. The method achieves state-of-the-art performance on the XD-Violence dataset in both unimodal and multimodal settings. Additionally, the method shows good effectiveness on the UCF-Crime dataset, further proving its strong generalization ability across datasets.

**Strengths:**

1.The proposed method is totally interesting and novel. It provides an effective solution to the ambiguous anomalies problem by progressively learning event context in hyperbolic spaces.

2.The proposed method achieves good performance, especially on XD-Violence dataset in both unimodal and multimodal settings. Visualization experiments also highlight its effectiveness in solving the ambiguous anomalies problem.

3.The HE-GCN and DSI modules are well-motivated, and the authors have reported sufficient ablation study results to prove the effectiveness of each module. I think it is much helpful.

**Weaknesses:**

1.For the HE-GCN module, I am curious about the relationship between HDE and LSHAD. The authors should explain the relationship between HDE and LSHAD in detail?

2.Figure 2 should be revised to clarify that the method is applicable to both unimodal and multimodal inputs, not just multimodal. Currently, it may mislead readers.

3.The Preliminaries section is well-detailed and informative, but it may be complex and difficult for a broader audience to understand.

4.There are some minor writing issues: Lines 439 and 479, “Figure” and “Fig” should be standardized; Line 147, there is a symbol display error.

**Questions:**

Same as weakness.

---

> ### Author Rebuttal · Authors · 2024-08-07
>
> ***Q1: For the HE-GCN module, I am curious about the relationship between HDE and LSHAD. The authors should explain the relationship between HDE and LSHAD in detail?***
>
> Thank you for the valuable comments. We will elaborate on the relationship between *HDE* and *LSHAD* in detail.
> In fact, *HDE* was designed to serve *LSHAD*. To enhance the model's ability to capture contextual information in hyperbolic space, we propose *LSHAD* as a threshold in the node selection for message aggregation, which ensures that the model first captures the broader global context with a relaxed threshold at the beginning of message aggregation, then gradually shifts focus to the local context with stricter thresholds.
> When there is a significant difference in features between nodes, we need to adopt a more relaxed node selection strategy to choose as many nodes as possible. However, as the number of layers increases and the features between nodes become more similar, we need to adopt a stricter selection strategy to choose the more important nodes.
> Therefore, we make the *LSHAD* relate to the degree of message aggregation of the current layer, which is measured by *HDE*. *HDE* essentially captures how effectively the nodes' information is aggregated and how similar their features become through the message-passing process.
> The synergy between *HDE* and *LSHAD* ensures that HE-GCN module effectively captures hierarchical relationships and aggregates information, enhancing the discriminative power of DSRL. By employing these strategies, our DSRL leverages the strengths of both Euclidean and hyperbolic geometries to improve the performance of VVD, particularly in distinguishing ambiguous violent events.
>
> ***Q2: Figure 2 should be revised to clarify that the method is applicable to both unimodal and multimodal inputs, not just multimodal. Currently, it may mislead readers.***
>
> We thank the reviewer for the valuable suggestions. We will modify Figure 2 to clarify that our method is applicable to both unimodal and multimodal inputs in the final version.
>
> ***Q3: The Preliminaries section is well-detailed and informative, but it may be complex and difficult for a broader audience to understand.***
>
> Thank you very much for your valuable feedback. We understand that the Preliminaries section may be complex for a broader audience, and we are committed to making it more accessible. In the final version of the paper, we will add further explanations for some of the equations, such as providing additional descriptions for Eq. 4 and Eq. 5.
> We have revised the explanation regarding the connections between hyperbolic space and tangent space. The original sentence:
> "The connections between hyperbolic space and tangent space are established by the exponential map $\exp _{\mathbf{x}}^{K}(\cdot)$ and logarithmic map $\log _{\mathbf{x}}^{K}(\cdot)$ are given as follows:" has been modified for clarity.
>
> The revised explanation is: "The mapping between hyperbolic spaces and tangent spaces can be done by exponential map and logarithmic map. The exponential map is a map from a subset of a tangent space of $\mathbb{L} _ {K}^{n}$ (i.e., $\mathcal{T} _ {\mathbf{x}} \mathbb{L} _ {K}^{n}$) to  $\mathbb{L} _ {K}^{n}$ itself. The logarithmic map is the reverse map that maps back to the tangent space. For points $\textbf{x} ,\textbf{y}  \in \mathbb{L} _ {K}^{n} $,  $\textbf{v}\in \mathcal{T} _ {\mathbf{x}} \mathbb{L} _ {K}^{n}$, such that $\textbf{v}\ne \textbf{0} $ and $\textbf{x}\ne \textbf{y} $, the exponential map $\exp _{\mathbf{x}}^{K}(\cdot)$ and logarithmic map $ \log _{\mathbf{x}}^{K}(\cdot) $ are given as follows:''.
>
> ***Q4: There are some minor writing issues: Lines 439 and 479, “Figure” and “Fig” should be standardized; Line 147, there is a symbol display error.***
>
> Thank you for your valuable comments. We will address these minor writing issues in the final version by changing "Fig" to "Figure" and correcting the symbol errors. Furthermore, we have carefully reviewed the paper to ensure that there are no writing errors.

---

> ### Comment · Reviewer_5VkS · 2024-08-08
> **new comments**
>
> The authors have addressed all my concerns, so I stay with my original score.

---

### Official Review · Reviewer_rxrS · 2024-07-09

**Soundness:** 3
**Presentation:** 3
**Contribution:** 3
**Rating:** 6
**Confidence:** 4

**Summary:**

This paper proposes leveraging dual-space learning, encompassing both Euclidean and Hyperbolic spaces, to enhance discriminative capacity by capitalizing on the strengths inherent in hyperbolic learning. To overcome the limitations of the hard node strategy in the previous method, this work introduces DSRL (Dual-Space Representation Learning), which enhances node aggregation selection by utilizing layer-sensitive hyperbolic association degrees constrained by hyperbolic Dirichlet energy. A cross-space attention mechanism is then proposed to facilitate information interactions between Euclidean and hyperbolic space to capture better discriminative features.

**Strengths:**

### Novelty:
Hyperbolic space learning is a valuable direction in the field of video understanding because there are inherent hierarchical relationships under video series. The previous method (HyperVD) is a good attempt, but it simply transfers the XD-Violence baseline into hyperbolic space without solving the limitations in hyperbolic GCN. The authors introduce a dual-space approach to combine the strengths of both Euclidean and Hyperbolic spaces to maximize discrimination capability and use hyperbolic Dirichlet energy to address the over-smoothing issue underlining the previous hyperbolic networks.
### Clarity:
Overall, the main paper is easy to follow and organized well.
### Experiments:
The main quantitative and qualitative experiments are adequate for the violence detection task, including ablation studies and t-SNE visualizations. The results demonstrate significant improvements compared to previous methods.

**Weaknesses:**

### Experiments:
It would have made the manuscript more convincing if the authors could provide inference visualizations of the ablation modules, which means to better illustrate the discriminative power of the method w/ or w/o your core modules.

### Minor Typos:
- Line 137:  $\mathcal{L}^{\/}$ should be $\mathcal{L}^{n}$.
- Eq (17) misses the right bracket for the $softmax$ function.
- In Eq (9), it misses the symbol annotations for $\mathbf{v}$. Did you use the hyperbolic transformation strategy proposed in "Fully hyperbolic neural networks", if so, the citation is missed here.

**Questions:**

- Why utilize these two spaces (hyperbolic and Euclidean) for the representation learning? Have you explored different hyperbolic spaces such as Lorentz and Poincaré Ball?
- How about the model's training stability?

**Limitations:**

Please refer to the weakness.

---

> ### Author Rebuttal · Authors · 2024-08-07
>
> Thank you so much for acknowledging the strength of our method. We have carefully considered your constructive and insightful comments and here are the answers to your concerns.
>
> ***Q1: It would have made the manuscript more convincing if the authors could provide inference visualizations of the ablation modules, which means to better illustrate the discriminative power of the method w/ or w/o your core modules.***
>
> Thank you for your positive comments and valuable suggestions. We have supplemented inference visualization results for the ablation module, which will be added in the final version. We conducted corresponding visualisation experiments to explore the discriminative power of the model w/ or w/o our core modules, HE-GCN and DSI. **Specific visualisations and analysis are shown in Figures 1 and 2 of the uploaded PDF file.** Two dimensions (the feature-level and the frame-leve) are analyzed in the two figures.
> **At the feature-level,** the results are shown in Figure 1 of the uploaded PDF. Compared to GCN, HE-GCN can capture the hierarchical context of events, effectively separating features. This results in a greater distance between feature clusters compared to Figure 1(a) original features and Figure 1(b) using only Euclidean representation learning. However, some challenging feature points remain difficult to distinguish. The addition of the DSI module facilitates information interactions between Euclidean and hyperbolic spaces, capturing more discriminative features to better differentiate these challenging feature points. As shown in Figure 1(d), the DSI module further enhances feature differentiation by effectively combining information from both spaces.
> Meanwhile, **at the frame-level**, experiments conducted on two test videos as shown in Figure 2 of the uploaded PDF demonstrate that our method significantly improves the discriminative power for identifying violent frames compared to the baseline, which uses only GCN. Compared with the model w/o our core modules, both HE-GCN and DSI contribute to detecting violent frames.
> Following your suggestions, we believe that we can more comprehensively demonstrate the effectiveness of our method and make our manuscript more convincing.
>
> ***Q2: Minor Typos:***
>
> Thank you for spotting these minor typos. We will modify them accordingly in the final version. In Eq(9), $\textbf{v} \in \mathbb{R}^{n+1}$ denotes a velocity (ratio to the speed of light) in the Lorentz transformations. And we used the hyperbolic transformation from this paper, which we will cite in the final version.
>
> ***Q3: Why utilize these two spaces (hyperbolic and Euclidean) for the representation learning? Have you explored different hyperbolic spaces such as Lorentz and Poincaré Ball?***
>
> Since we focus on addressing ambiguous violence in the VVD task, it is essential to consider both visual features and the hierarchical contextual information of the event. Therefore, we employ these two spaces for representation learning for two main reasons:
> 1) Euclidean space is widely used in many domains and is effective at capturing visual features, such as salient motion and shape changes in videos. However, it often overlooks the relationships between events.
> 2) Hyperbolic space, characterized by exponentially increasing metric distances, naturally reflects the hierarchical structure of data. It enhances the hierarchical relationships of events but tends to weaken the expression of visual features.
>
> By combining these two spaces for representation learning, we can leverage their respective strengths to improve the discriminative of feature representations, ultimately enhancing the performance of VVD.
>
> In the preliminary practice, we implement our method with the Poincaré ball model.  However, we encountered situations where the loss became NaN, which is caused by the numerical instability of the Poincaré ball model. Subsequently, we switched to the Lorentz model because it guarantees numerical stability and computational simplicity in its exponential and logarithmic maps and distance functions. Due to the stable training and improved performance, we finally selected the Lorentz model during hyperbolic representation learning.
>
> ***Q4: How about the model's training stability?***
>
> We evaluate the training stability of our model by analyzing the trends in training loss and average precision (AP). **The results presented in Figure 3 of the uploaded PDF file** exhibit a stable training process.
>
> **Training loss curve over time.** The training loss starts at approximately 0.61 and gradually decreases over time, stabilizing around 0.17 towards the end of the training. This consistent decrease and eventual stabilization indicate effective learning and convergence throughout the training process.
> **Average precision curve over time.** The AP shows some fluctuations during the early epochs, which is normal as the model adjusts its weights and parameters. Despite these fluctuations, the overall trend is upward, indicating that the model's performance is improving. Subsequently, the AP values show less variation and remain at a high level.

---

> > ### Comment · Reviewer_rxrS · 2024-08-10
> > **Comments from Reviewer rxrS**
> >
> > Thank you for the authors' detailed response, most of my concerns have been addressed and I maintain my original rating.

---

### Official Review · Reviewer_8atP · 2024-07-11

**Soundness:** 3
**Presentation:** 3
**Contribution:** 2
**Rating:** 5
**Confidence:** 4

**Summary:**

The paper introduces a novel method called Dual-Space Representation Learning (DSRL) aimed at enhancing the detection of video violence, particularly in scenarios where the violence is weakly supervised and visually ambiguous.

DSRL combines the strengths of both Euclidean and hyperbolic geometries to capture discriminative features for violence detection, leveraging the hierarchical structure modeling capability of hyperbolic spaces.

Two specialized modules are designed: the Hyperbolic Energy-constrained Graph Convolutional Network (HE-GCN) module and the Dual-Space Interaction (DSI) module, enhancing the understanding of event hierarchies and promoting dual-space cooperation.

Comprehensive experiments on the XD-Violence dataset demonstrate the effectiveness of DSRL, outperforming existing methods in both unimodal and multimodal settings.

**Strengths:**

1. The method is innovative. DSRL designs a special information aggregation strategy. Through layer-sensitive hyperbolic association degree (LSHAD) and hyperbolic Dirichlet energy (HDE), it effectively captures the hierarchical context of events and can better model the problem of violent video detection.

2. This method uses the property of hyperbolic space that can better distinguish visually similar events and is applied to a suitable scenario - fuzzy violent event recognition, which improves recognition accuracy.

3. The performance of the method is outstanding. It has reached SOTA in quantitative analysis. In qualitative analysis, DSRL demonstrates its ability to distinguish between violent and normal events in different situations, including its performance in complex violent events.

4. The proposed model can process multimodal inputs, integrate visual and audio information, and improve the ability to understand complex scenes.

**Weaknesses:**

Subjectivity: The Layer-Sensitive Hyperbolic Association Degree (LSHAD) proposed by the author contains multiple hyperparameters, and LSHAD is used to examine the threshold of the message graph. It is very important. There is a lack of explanation for why it is designed in this way.

Complexity: DSRL combines Euclidean and hyperbolic geometric spaces, as well as cross-space interaction mechanisms, which may increase the complexity of the model, resulting in higher demands on computing resources and training time.

Parameter sensitivity: DSRL contains multiple hyper-parameters, such as β, γ, α, etc. The selection of these parameters may have a significant impact on model performance. This article lacks an analysis of parameter adjustment.

Application-specific limitations: DSRL is specifically designed for the task of video violence detection, which may mean that further adjustments or optimizations are required when applying it to other types of video content analysis tasks.

**Questions:**

1. What are the specific reasons for the design choices in the Layer-Sensitive Hyperbolic Association Degree (LSHAD) with its multiple hyperparameters and threshold criteria? Was there any experimental or theoretical analysis conducted to support these design decisions?
2. Considering the increased demand for computing resources and training time, are the benefits of these complex designs justified?
3. What methods or heuristic approaches are used to determine the optimal values of these hyperparameters?  Has a systematic sensitivity analysis been conducted for the selection of hyperparameters (such as β, γ, α, etc.) in the DSRL model?
4. Has the DSRL model been tested on other types of video content analysis tasks? If so, how did the model perform in these tasks? Are there any additional adjustments or optimizations required to adapt the model to different video content analysis applications?

**Limitations:**

The authors discussed their limitations.

---

> ### Author Rebuttal · Authors · 2024-08-07
>
> We thank the reviewer for the comments and suggestions to improve this work. We will address your concerns below.
>
> ***Q1: Reasons for the design choices in the LSHAD with its multiple hyperparameters and threshold criteria***
>
> Inspired by the Global-first principle [1] that humans always have cognition on global first and then focus on local, we propose a novel node selection strategy, which guarantees the model captures the broader global context first with a relaxed threshold at the beginning of message aggregation and then focuses on the local context with more strict thresholds.
> To achieve this, we introduce the *LSHAD* construction rule, which calculates an *LSHAD* threshold based on the number of the current layer *k* and hyperbolic Dirichlet energy of the current layer. As the *k* increases and the hyperbolic Dirichlet energy decreases, the *LSHAD* threshold increases and is limited to [0,1] by the sigmoid function. If there is no $\beta$ and $\gamma$, the threshold in the first layer will be strict ($\textgreater 0.5$), causing the overlook of some global context information. Therefore, to make our node selection threshold conform to the Global-first principle, we introduced the two hyperparameters in *LSHAD*, where $\beta$ controls the influence of the number of current layer *k* and $\gamma$ acts as a bias to fine-tune the threshold. Moreover, we conducted an ablation study to determine the optimal value of the two hyperparameters ($\beta$,$\gamma$), where $\beta$ ranges among [0.2,0.4,0.6,0.8,1.0] and $\gamma$ ranges among [1.0,1.2,1.4,1.6,1.8,2.0]. The results in the table below reveal that when $(\gamma - \beta)$ is 0.4, the performance is optimal, so we chose a pair (0.8, 1.2) from this set.
> | $\beta \setminus \gamma $ | 1.0  | 1.2  | 1.4  | 1.6  | 1.8  | 2.0  |
> |--------------------------|------|------|------|------|------|------|
> | **0.2**                      | 85.22| 87.12| 86.32| 86.60 | 86.31| 86.86|
> | **0.4**                      | 87.52| 85.22| 87.12| 86.32| 86.60| 86.31|
> | **0.6**                      | 87.61| 87.52| 85.22| 87.12| 86.32| 86.60 |
> | **0.8**                      | 87.29| 87.61| 87.52| 85.22| 87.12| 86.32|
> | **1.0**                      | 86.29| 87.29| 87.61| 87.52| 85.22| 87.12|
>
> [1] Chen, L. (1982). Topological structure in visual perception. Science, 218, 699-700.
>
> ***Q2: Comparisons of computing resources and training time***
>
> Our designs, HE-GCN and DSI, effectively capture hierarchical contextual information of events and integrate two spaces of information, respectively.
> Following your suggestion, we trained three models on XD-Violence for 30 epochs each using a single NVIDIA RTX A6000 GPU: Baseline (GCN), Baseline+HE-GCN, and Baseline+HE-GCN+DSI (DSRL). The results in the table below show that the benefits of our designs are justified. Our DSRL improved AP by 3.57\% compared to the Baseline, while the training time per epoch increased by only 41 seconds, and memory usage rose by just 4.1GB, both of which are within a reasonable range, making the performance gains well worth the resource consumption.
> | Methods                    | Params  | Training Time per Epoch | Total Training Time | Video Memory Usage | AP (%) |
> |----------------------------|---------|:-------------------------:|:---------------------:|:--------------------:|:--------:|
> | Baseline (GCN)             | 0.7734M | 2 min                   | 60 min              | 4.24 GB            | 84.04  |
> | Baseline + HE-GCN          | 0.8975M | 2 min 19 s              | 69 min 39 s         | 7.03 GB            | 86.46  |
> | Baseline + HE-GCN + DSI (DSRL) | 0.9966M | 2 min 41 s              | 80 min 19 s         | 8.34 GB            | 87.61  |
>
> ***Q3: Hyperparameter sensitivity analysis***
>
> We employed a grid search method to determine the optimal values of the hyperparameters in our model. We have conducted a sensitivity analysis on $\alpha$ and $\lambda$ in DSI, as shown in Line 470 in our submitted paper.
> We further conducted a sensitivity analysis on $\beta$ and $\gamma$. The table referenced in **Q1** shows that our model is relatively robust to changes in the hyperparameters within certain ranges.
> Besides, the table also shows that when $(\gamma - \beta)$  is 0.4, the performance is optimal, so we chose the pair (0.8, 1.2) from this set.
> Specifically, when $\beta$=0.8, the variations in $\gamma$ within the tested range do not significantly affect the model's performance. Similarly, when $\gamma$=1.2, changes in $\beta$  also have minimal impact on the model's effectiveness. This indicates that our model's performance is relatively stable under small perturbations of these hyperparameters, suggesting a degree of robustness in the parameter configuration.
>
> ***Q4: Applicability to other video content analysis tasks***
>
> Currently, we primarily focus on addressing ambiguous violence in VVD by customising DSRL based on the dual-space idea. Although we have not yet tested its effectiveness on other tasks, our theoretical analysis suggests that this dual-space idea could be beneficial for various other video content analysis tasks. For instance, in video person re-identification (Video Re-ID), capturing both visual features and spatio-temporal hierarchical relationships is crucial. To adapt our model to Video Re-ID, we may consider the following adjustments:
> 1) We may need to adjust the graph construction to fully incorporate information about human body parts and their hierarchical relationships.
> 2) Video Re-ID is a type of fine-grained recognition that may require enhancing visual feature extraction in Euclidean space to strengthen node features in the graph.
>
> In the future, we will explore adapting the model to different video content analysis applications.

---

> > ### Comment · Reviewer_8atP · 2024-08-14
> >
> > Thank you for the authors' response, most of my concerns have been addressed, and considering the application area would be somewhat domain specific, I will stay with my original rating, i.e. being slightly positive.

---

### Author Rebuttal · Authors · 2024-08-07

We sincerely thank all the reviewers for providing constructive feedback that helped us improve the paper. We are glad the reviewers find that

 **Our method is interesting and novel**

* "The method is innovative." ---8atP
* "The proposed method is totally interesting and novel." ---5VkS

* "This paper presents a pioneering approach." ---8SDB

**Our presentation is clear and well-motivated**

* "Overall, the main paper is easy to follow and organized well." ---rxrS

* "The HE-GCN and DSI modules are well-motivated." ---5VkS

* "The quality of this paper's presentation is good and the whole paper is well-organized"---8SDB

**Our performance is outstanding**

* "The performance of the method is outstanding. " ---8atP

* "The results demonstrate significant improvements compared to previous methods. " ---rxrS

*  "The proposed method achieves good performance " ---5VkS

*  "Experiments on XD-Violence and UCF-Crime datasets show the advantage of the proposed method. " ---8SDB

**Main contributions**

1. We present DSRL, the first method to integrate Euclidean and hyperbolic geometries for VVD, which significantly improves the discrimination of ambiguous violence and achieves state-of-the-art performance on the XD-Violence dataset in both unimodal and multimodal settings.

2. We design the HE-GCN module with a novel message aggregation strategy to better capture the hierarchical context of events, where the node selection threshold is dynamic, not fixed, and determined by layer-sensitive hyperbolic association degrees based on hyperbolic Dirichlet energy.

3. We design the DSI module to break the information cocoon for better dual-space cooperation, combining visual discrimination from Euclidean geometry and event hierarchical discrimination from hyperbolic geometry, utilizing cross-space attention to facilitate information interactions.

Once again, we express our sincere appreciation for your valuable contributions to the review process. Your expertise and guidance have been invaluable in improving the quality of our work. We remain committed to continuous improvement and eagerly await your final decision. Please see the attached PDF for a one-page PDF with added experimental results.

---

### Decision · Program_Chairs · 2024-09-25

**Decision:**

Accept (poster)

**Comment:**

The paper proposes a method for violence detection from videos. The reviewers found the method interesting, novel and effective. The evaluation shows the effectiveness of the proposed method. The authors rebuttal further clarified questions from the reviewers. On the other hand, it is very important to acknowledge and discuss the issue of potential model bias. For example, the learned model may work better just by taking advantage of false association between gender/race and violence. This has been shown in many related works, such as racial bias in hate speech detection. It is very strongly recommended that the authors discuss this issue in details and also perform additional analysis showing the model (or the improvement) is not biased against protected attributes.